

# Chitin distribution in the *Oithona* digestive and reproductive systems revealed by fluorescence microscopy

Kevin Sugier[1], Benoit Vacherie[2], Astrid Cornils[3], Patrick Wincker[1], Jean-Louis Jamet[4] and Mohammed-Amin Madoui[1]

[1] Génomique Métabolique, Genoscope, Institut François Jacob, CEA, CNRS, Univ Evry, Université Paris-Saclay, Evry, France
[2] Commissariat à l'Energie Atomique (CEA), Institut François Jacob, Genoscope, Evry, France
[3] Alfred-Wegener-Institut Helmholtz-Zentrum für Polar- und Meeresforschung, Polar Biological Oceanography, Bremerhaven, Germany
[4] Université de Toulon, Aix Marseille Université, CNRS/INSU, IRD, MIO UM 110, Mediterranean Institute of Oceanography, La Garde, France

Corresponding author
Kevin Sugier,
ksugier@genoscope.cns.fr

## ABSTRACT

Among copepods, which are the most abundant animals on Earth, the genus *Oithona* is described as one of the most numerous and plays a major role in the marine food chain and biogeochemical cycles, particularly through the excretion of chitin-coated fecal pellets. Despite the morphology of several *Oithona* species is well known, knowledge of its internal anatomy and chitin distribution is still limited. To answer this problem, *Oithona nana* and *O. similis* individuals were stained by Wheat Germ Agglutinin-Fluorescein IsoThioCyanate (WGA-FITC) and DiAmidino-2-PhenylIndole (DAPI) for fluorescence microscopy observations. The image analyses allowed a new description of the organization and chitin content of the digestive and reproductive systems of *Oithona* male and female. Chitin microfibrils were found all along the digestive system from the stomach to the hindgut with a higher concentration at the peritrophic membrane of the anterior midgut. Several midgut shrinkages were observed and proposed to be involved in faecal pellet shaping and motion. Amorphous chitin structures were also found to be a major component of the ducts and seminal vesicles and receptacles. The rapid staining protocol we proposed allowed a new insight into the *Oithona* internal anatomy and highlighted the role of chitin in the digestion and reproduction. This method could be applied to a wide range of copepods in order to perform comparative anatomy analyses.

## INTRODUCTION

Copepods are the most abundant animals on Earth ahead of insects and nematodes (*Humes, 1994*) and inhabit all aquatic niches: groundwater, vernal ponds, glaciers, lakes, rivers and oceans (*Huys & Boxshall, 1991*). Among marine copepods, *Oithona* has been described as the most important marine planktonic genus in terms of abundance (*Gallienne & Robins, 2001*). A recent study, based on the *Tara* Oceans metagenomic data,

has shown the global distribution of *Oithona* in coastal and open ocean waters (*Madoui et al., 2017*), which highlighted its key role as a major secondary producer of the marine food chain (*Beaugrand et al., 2003*; *Zamora-Terol et al., 2014*). The important contribution of copepods in the biological carbon pump has also been demonstrated (*Jonasdottir et al., 2015*), in particular through the excretion of faecal pellets (*Steinberg & Landry, 2017*) that sink, provide organic and inorganic compounds to microplankton (*Steinberg, Goldthwait & Hansell, 2002*; *Valdés et al., 2017*), and deposit on the sediments where they could remain as fossils for several thousand years (*Bathmann et al., 1987*; *Haberyan, 1985*). The biochemical analysis of the copepod faecal pellets has revealed a high amount of chitin (*Kirchner, 1995*), a β-1-4-*N*-acetylglucosamine polymer, the most abundant biopolymer in nature after celluloses (*Kirchner, 1995*), and mostly known in copepods as a component of the exoskeleton. Besides the role of copepods in the carbon pump, the abundance of chitin in the faecal pellets also points out the implication of copepods in the global nitrogen cycle (*Frangoulis, Christou & Hecq, 2004*).

Morphological traits of more than forty *Oithona* species are well known (*Razouls et al., 2005–2018*), especially the structure of the antennules, the oral appendages, the swimming legs and the caudal rami (*Nishida, 1985*). However, such morphological traits are only accessible through finical dissections under the microscope that need expertise and are time-consuming. Recently, molecular tools have proven their usefulness in species identification (*Cornils, Wend-Heckmann & Held, 2017*; *Madoui et al., 2017*).

The detailed external anatomy of copepods has been analysed through Congo red fluorescence (*Michels & Büntzow, 2010*) and through electron microscopy that allowed the species identification of copepods and the characterization of their external structures (*Chang, 2013*; *Cuoc et al., 1997*; *Marques et al., 2017*). Using an electron microscope, *Oithona nana* Giesbrecht, 1892 female sexual orifices with attached male spermatophores were observable (*Huys & Boxshall, 1991*). Diagrams of marine and freshwater cyclopoids, which provide the structures of the reproductive and digestive systems (*Borradaile & Potts, 1935*; *Dussart & Defaye, 2001*; *Kellogg, 1902*) were available. Some studies proposed methods to observe the reproductive system of aldehyde-preserved copepods by direct light microscopy observation of individuals (*Eisfeld & Niehoff, 2007*; *Niehoff, 2003*; *Niehoff & Hirche, 1996*; *Tande & Hopkins, 1981*), by staining of gonad with borax carmine (*Tande & Gronvik, 1983*; *Tande & Hopkins, 1981*), with fluorescent polyunsaturated aldehydes (PUAs) probes (*Wolfram, Nejstgaard & Pohnert, 2014*), or with Fast Green (*Batchelder, 1986*). The internal anatomy of *O. similis* Claus, 1863 has been recently described using phase contrast microscopy and provided the first insight into the organization of the *Oithona* female reproductive system (*Mironova & Pasternak, 2017*). In the *Wolfram, Nejstgaard & Pohnert (2014)* study, some pictures of the calanoid *Acartia tonsa* obtained using fluorescent PUA probes, also allowed to determine the anatomy of the digestive system. Other studies (*Bautista & Harris, 1992*; *Debes, Eliasen & Gaard, 2008*) used the chlorophyll fluorescence to determine the ingestion rates and the gut contents, but without providing a clear structure of the digestive organs. Electron microscopy revealed that chitin microfibrils are present in the anterior and posterior midgut peritrophic membrane (PM) of free-living and in the

posterior PM of parasitic copepods (*Yoshikoshi & Kô, 1988*), but no *Oithona* species have been included in the study.

For a better understanding of the ecological success of *Oithona*, a detailed knowledge of its internal anatomy is crucial. Fluorescence microscopy based on a double staining coupling Wheat Germ Agglutinin-Fluorescein IsoThioCyanate (WGA-FITC) and Diamidino-2-phenylIndole (DAPI) were used to elucidate the internal anatomy. DAPI is a blue fluorescent protein which has an affinity to two nucleoids: adenine and thyrosin (*Lin, Comings & Alfi, 1977*). This staining is widely used to detect DNA in eukaryotes, prokaryotes and some viruses, without tissue-specificity. FITC is a green fluorescent protein that can be conjugated with a wheat lectin that has an affinity and specificity to *N*-acetyl-β-D-glucosamine (*Allen, Neuberger & Sharon, 1973*). WGA-FITC staining is widely used for chitin detection by fluorescence, in a liquid medium containing lysed cells or directly on whole organisms (*El Gueddari et al., 2002*; *Farnesi et al., 2015*; *Fones, Mardon & Gurr, 2016*; *Godoy, Fernandes & Martins, 2015*). On copepods, WGA-FITC was used only once; but after dissolution of the soft tissues which did not allow the investigation of the internal anatomy (*Mravec et al., 2014*). In the present study, we used WGA-FITC and DAPI staining to provide a new insight into the internal anatomy and chitin content of *O. nana* and *O. similis* with a focus on their digestive and reproductive systems.

## MATERIAL AND METHODS

### Biological materials samples

*Oithona nana* and *O. similis* specimens were sampled at two locations of the Toulon harbour, France, at the East of the little harbour of Toulon (Lat 43°06′52.1″N and Long 05°55′42.7″E) and the North of the great harbour of Toulon (Lat 43°06′02.3″N and Long 05°56′53.4″E). Sampling took place in November 2016, January, March and June 2017. The samples were collected from the upper water layers (0–10 m) using zooplankton nets with a mesh of 90 and 200 μm. Samples were preserved in 70% ethanol and stored at −4 °C. In the samples, individuals of the four different development stages were observable (nauplii, copepodites and adults of both sexes), but the large majority were female adults.

### Individual staining

This protocol was adapted from *Farnesi et al. (2015)*. After gently mixing the ethanol preserved samples (about 20 reversals), 100 μL were sampled in a 1.5 mL tube. After 2 min, the ethanol was removed, and 100 μL of phosphate buffered saline (PBS) at 1× and 10 μL of WGA-FITC at 2 mg mL$^{-1}$ (L4895 SIGMA, Lectin from Triticum vulgaris (wheat) FITC conjugate, lyophilized powder; Sigma-Aldrich, St. Louis, MO, USA) were added for chitin staining. After mixing, the sample was incubated for 30 min protected from light before supernatant removing. To stain the DNA, dual staining with DAPI can be performed by adding, 100 μL of PBS at 1× and 10 μL of DAPI (D9542 SIGMA, DAPI for nucleic acid staining; Sigma-Aldrich, St. Louis, MO, USA) at 10×. The microscopy observations were done directly after mixing. This protocol can also be used on living individuals from a seawater sample; in this case, sodium chloride at 39 g L$^{-1}$ has to be added to the PBS solution.

## Microscopy

The stained individuals were placed between slide and coverslip and observed under a reflected fluorescence microscope Olympus BX43. WGA-FITC was excited with the 460/495 nm line from a 100 W mercury lamp with an interference excitation filter (BP460), and collected with a 505 nm dichroic mirror (DM505) and a 510 nm interference barrier filter (BA510IF). DAPI fluorescence was excited with the 340/390 nm line from a 100 W mercury lamp with an interference excitation filter (BP340), and collected with a 410 nm dichroic mirror (DM410) and a 420 nm interference barrier filter (BA420IF). Selected *Oithona* individuals were photographed with a 16-megapixel camera using the ToupView software (v.3.7). For each individual, three photographs were taken: one in polarized light, one with the WGA-FITC fluorescence and one with the DAPI fluorescence. Some colour adjustments were made with the ImageJ software (*Schneider, Rasband & Eliceiri, 2012*).

# RESULTS

## *Oithona* morphology with WGA-FITC microscopy

The *Oithona* chitin was labelled with WGA-FITC directly on the individuals and observed by fluorescence microscopy. The setae and spines of the exopod segments of the five leg pairs could be identified and counted on *O. nana* (Fig. 1A). These first results revealed the chitinous structure of the setae and the spines, and could provide a rapid method for taxonomical identification. However, because of the individuals and setae position on the plate, we were not able to identify and count the setae of all tested individuals. Chitinous elliptic or spherical structures of unknown function and larger than 6 μm (Fig. 1A) were also visible in the exopods of the swimming legs. These globular structures were observed in both sexes of *O. nana* (Figs. 1A and 2A–2E), but only in female individuals of *O. similis* (Figs. 2F and 2G). They may also be smaller (Fig. 2F), or absent (Figs. 3A and 4C) in other individuals. These structures can also be present in other exopod segments (Fig. 2A). Another tubular structure, in the distal part of the exopods three of the right third leg, right and left fourth legs and right and left fifth legs were noticeable (Fig. 1A). In other *Oithona* individuals, these tubular structures appear to be attached to the globular structure (Figs. 2B, 2D and 2G).

## Chitin distribution in the *Oithona* digestive system

Chitin was detected all along the digestive system, from the stomach to the hindgut of the nauplius (Fig. 1B) and adults (Figs. 1C–1E) of the two species. The exoskeleton chitin was also stained by the WGA-FITC, which allowed a clear identification of the stomach in the prosome, of the midgut in the prosome and in the urosome and the hindgut in the urosome. Along the digestive system, the chitin had a microfibrillar structure aligned along the antero-posterior axis with regions showing higher microfibrils density, especially the anterior midgut and some stomach areas (Figs. 1C–1F). Some individuals contained in their anterior and posterior midgut one or several elliptical faecal pellets completely engulfed by chitin (Figs. 1E and 4B). However, no faecal pellets were found in the nauplius. In the anterior and posterior midgut, we observed several shrinkages

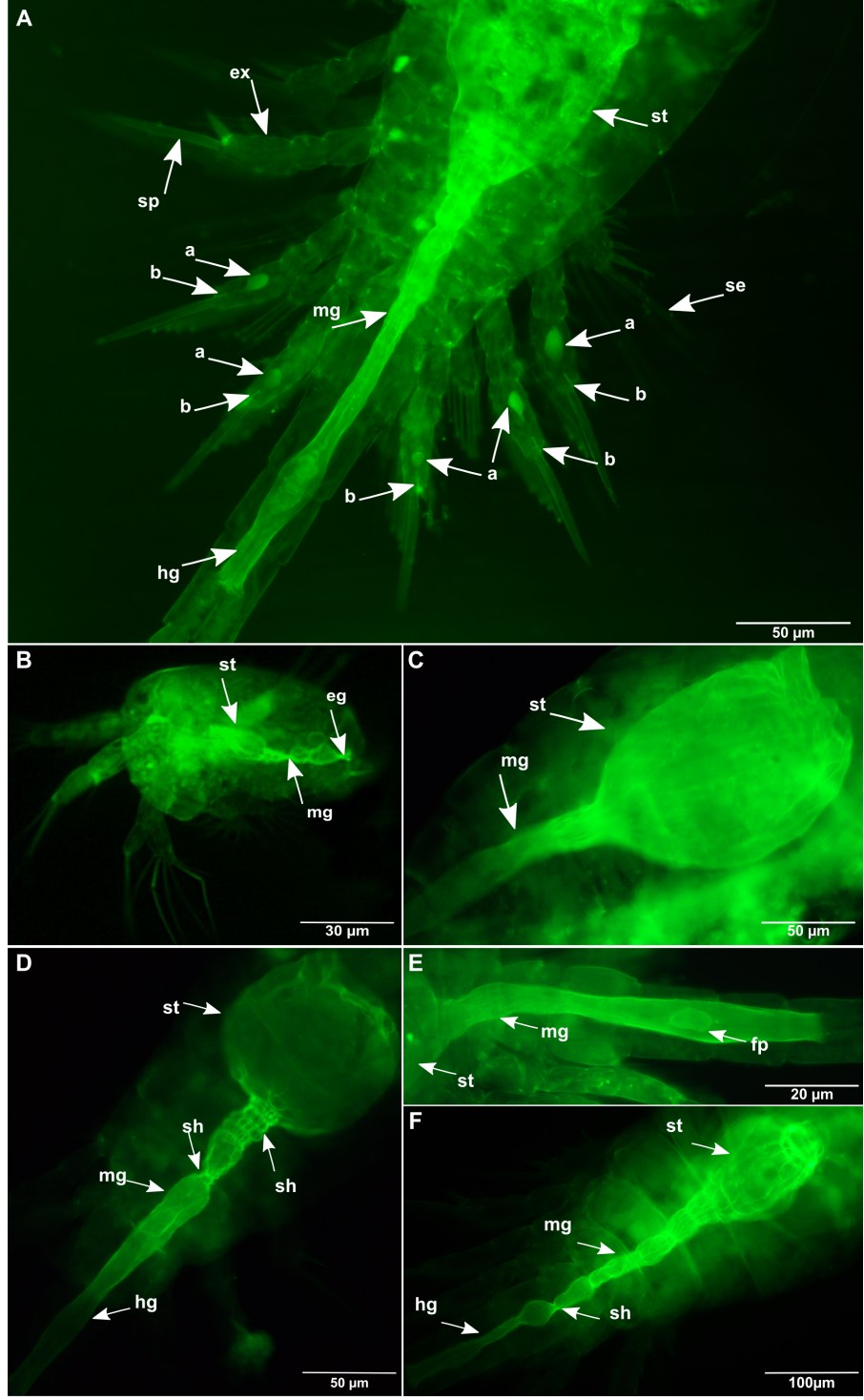

**Figure 1** *Oithona* **appendages morphology and digestive system by WGA-FITC fluorescence microscopy.** (A) Dorsal view of the *O. nana* female swimming appendages. (B) Lateral view of the *Oithona* nauplius digestive system. (C) Lateral view of the *O. nana* female stomach. (D) Dorsal view of the *O. nana* female stomach. (E) Lateral view of an *O. nana* male gut. (F) Dorsal view of an *O. similis* female adult stomach. st, stomach; mg, midgut; hg, hindgut; sh, shrinkage; ex, exopod; se, seta; sp, spine; fp, faecal pellet; a, globular structure; b, tubular structure.

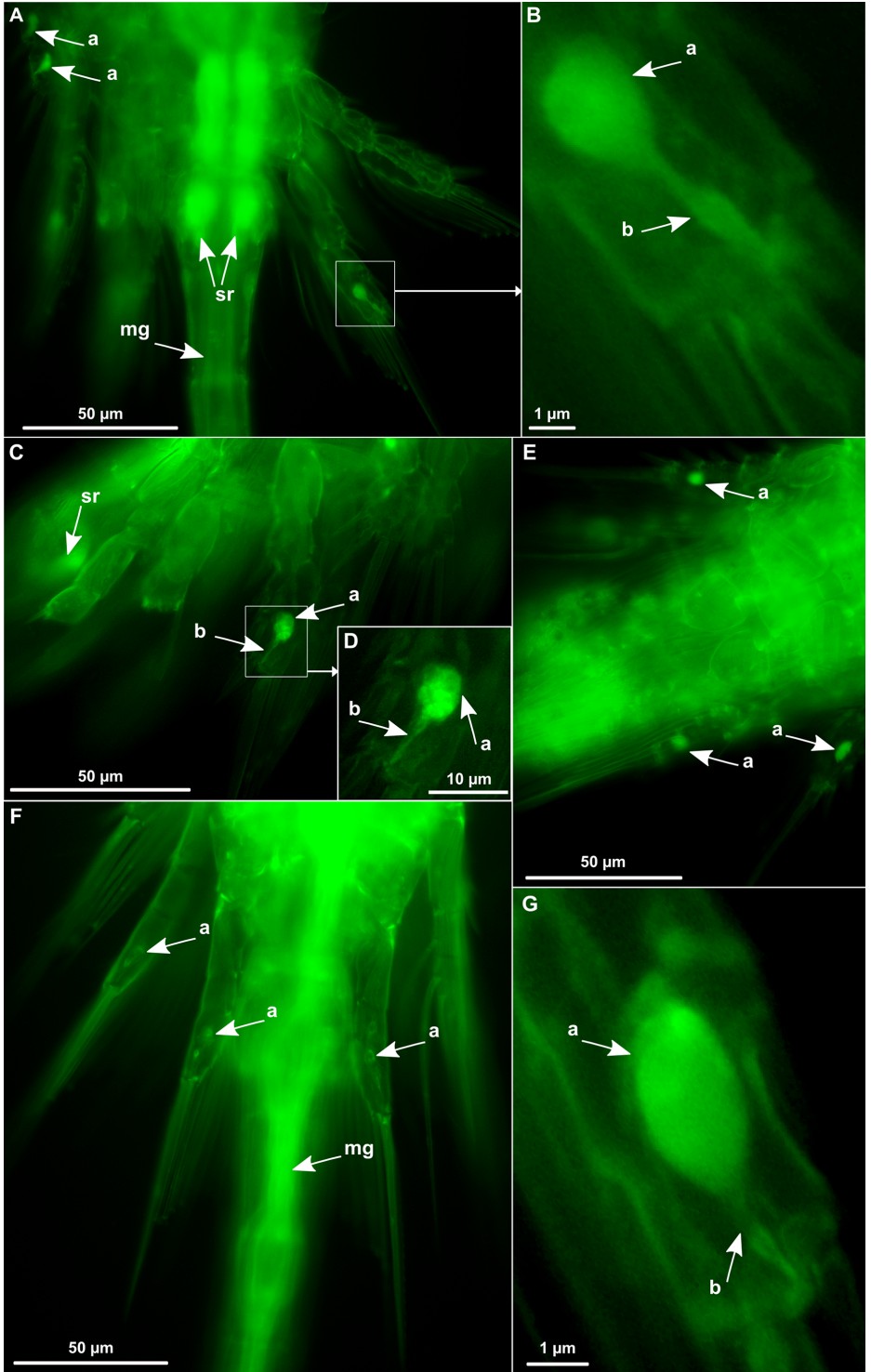

**Figure 2** *Oithona* **globular and tubular chitinous structures in the swimming appendages by WGA-FITC fluorescence microscopy.** (A) Dorsal view of *O. nana* female urosome/prosome junction. (B) Zoom on the right P4 exopod of the same *O. nana* individual. (C) Lateral view of *O. nana* female swimming appendages. (D) Zoom on the right P3 exopod of the same *O. nana* individual (E) Ventral view of *O. nana* male abdomen. (F) Dorsal view of *O. similis* female urosome/prosome junction. (G) Zoom on the right P5 exopod of a female *O. similis*. sr, seminal receptacle; mg, midgut; a, globular structure; b, tubular structure.

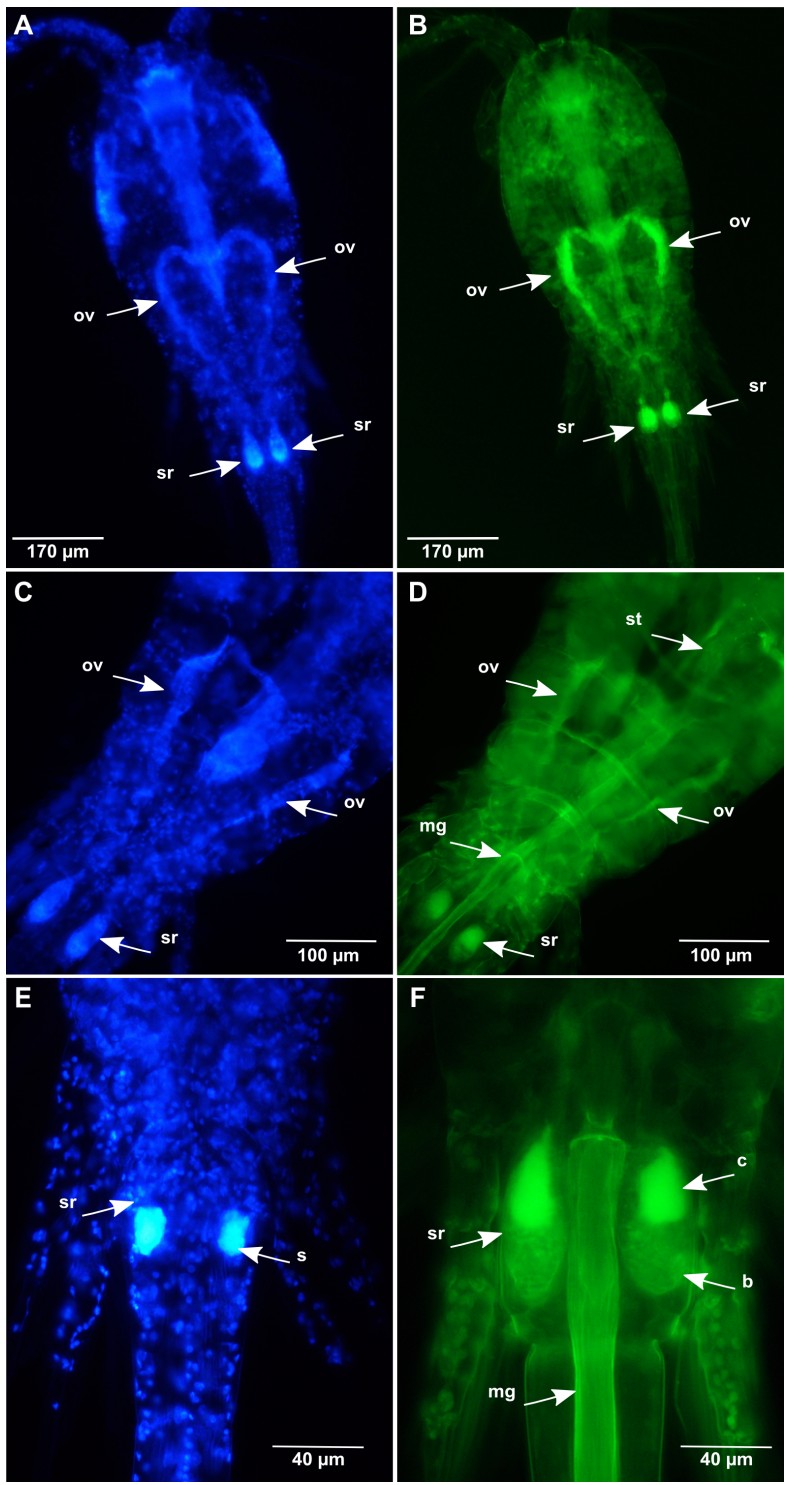

**Figure 3 *Oithona* female reproductive system by DAPI and WGA-FITC fluorescence microscopy.**
(A and B) Dorsal view of the *O. nana* female reproductive system (DAPI staining on the left and WGA-FITC staining on the right). (C and D) Dorsal view of the *O. similis* female reproductive system (DAPI staining on the left and WGA-FITC staining on the right). (E and F) Dorsal view of the *O. nana* female double sexual somite (DAPI staining on the left and WGA-FITC staining on the right). mg, midgut; sr, seminal receptacle; s, semen; ov, oviduct; hg, hindgut; b, diffuse chitin region; c, chitin rich region.

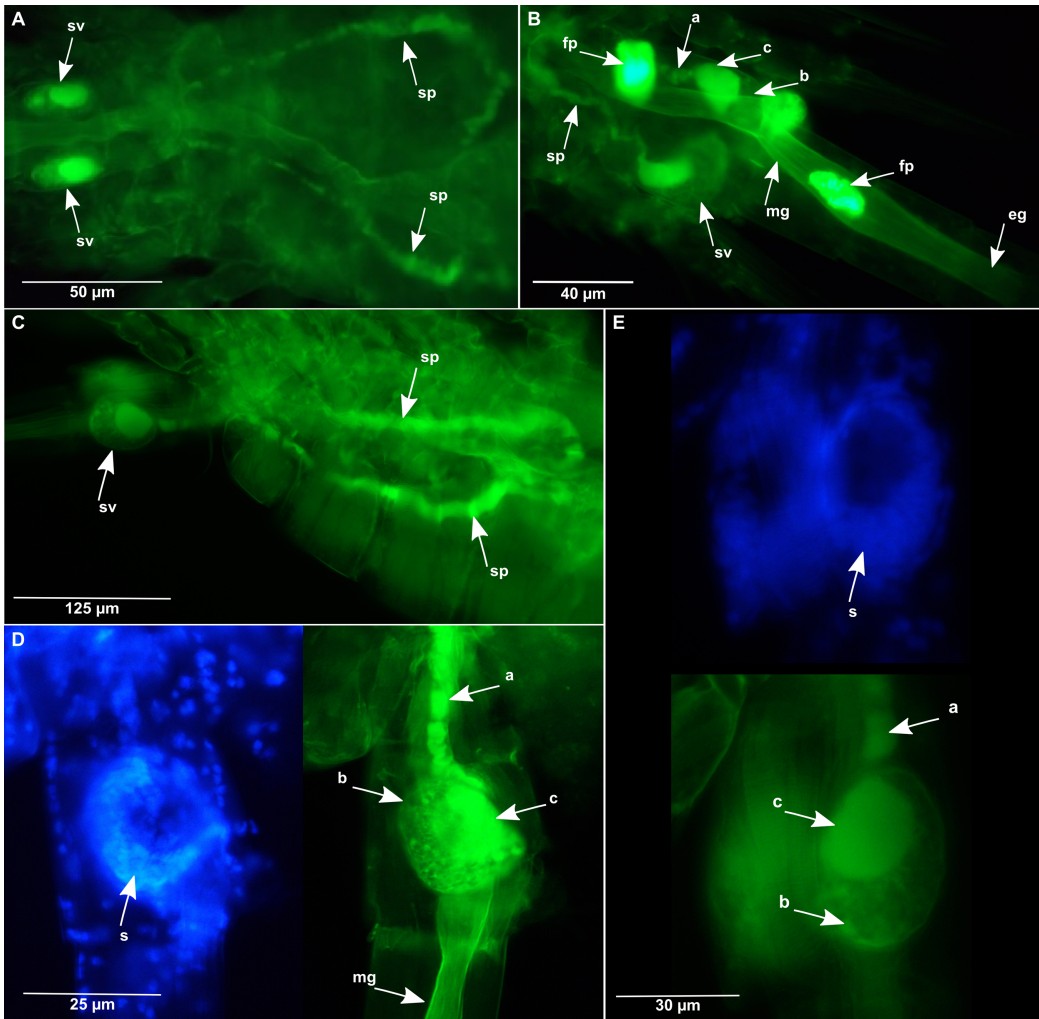

**Figure 4 *Oithona* male reproductive system by DAPI and WGA-FITC fluorescence microscopy.** (A) Dorsal view of the *O. nana* male reproductive system. (B) Dorso-lateral view of the *O. nana* seminal vesicle. (C) Lateral view of the male *O. similis* reproductive system. (D) Lateral view of the *O. nana* male double sexual somite. (E) Lateral view of the *O. similis* male double sexual somite. mg, midgut; fp, faecal pellet; sv, seminal vesicle; sp, spermiduct; a, heterogeneous chitin; b, diffuse chitin; c, chitin rich region.

at different interval distances corresponding to midgut contractions. In certain cases, several shrinkages (up to four) were separated by less than 5 μm (Fig. 1D), while other individuals showed more distant shrinkages (Fig. 1F).

## Chitin distribution in the *Oithona* reproductive system

The DAPI and WGA-FITC stainings on *Oithona* females allowed the identification of the ovaries and the oviducts that presented a heart shape in the middle of the prosome (Figs. 3A and 3B) as previously described by Mironova and Pasternak. The oviducts start from each lateral side of the gonads to the seminal receptacle in the genital double somite (the first two segments of the urosome). Comparing to the microfibrillar structure of the chitin found in the digestive system, the chitin staining in the reproductive

Sugier et al. (2018), *PeerJ*, DOI 10.7717/peerj.4685  8/15

system was mainly amorphous. Besides, its distribution was discontinuous along the ducts, altering chitin-rich and poor areas (Figs. 3C and 3D).

In females, we distinguished two parts forming the seminal receptacle (Figs. 3E and 3F). The first part was chitin-rich and located in the anterior region of the receptacle. The chitin distribution between the anterior receptacle and the oviduct was discontinuous. The second part was located in the posterior receptacle and contained less and sparser chitin, presenting a mix of microfibrillar and amorphous structures. Thanks to DAPI staining, in some females the presence of the DNA rich material in the posterior region of the seminal receptacle was observed and was likely to be male semen.

In males, the chitin staining allowed the identification of the spermiducts, which presented the same chitin pattern observed in the oviducts (Figs. 4A and 4C). The spermiducts probably start from each side of the male gonads (not visible on the pictures) to the seminal vesicles, in the sexual somite (Fig. 4B). As for the female seminal receptacle, the male seminal vesicle can be divided into two parts (Figs. 4D and 4E). The first part of the vesicles is chitin-rich, located in the anterior region of the vesicle. The distribution of the chitin from this upper part of the vesicle to the spermiduct was not continuous. The second part, located in the posterior region of the vesicle, was observed by DAPI staining and was likely to be filled by DNA-rich male semen.

## DISCUSSION

Comparing to previous staining methods used to observe the digestive and reproductive systems of copepods (*Batchelder, 1986*; *Eisfeld & Niehoff, 2007*; *Mironova & Pasternak, 2017*; *Niehoff, 2003*; *Niehoff & Hirche, 1996*; *Tande & Gronvik, 1983*; *Tande & Hopkins, 1981*; *Wolfram, Nejstgaard & Pohnert, 2014*), the protocol proposed here allows a clear insight into the chitin distribution in these systems. Moreover, this protocol is simple and rapid, taking a few minutes of manipulation, 30 min of incubation, and can be used on living, but also on alcohol-preserved copepods. The main limit of our method remains in the short-time staining of the WGA-FITC: a picture must be taken, a few minutes after fluorescence excitation to save any microscopic observation without loss of quality. Furthermore, for the reproductive system, the DAPI staining allows only the observation of the gonad structure; while the Mironova and Pasternak protocol allows a better identification of the oocytes.

The use of WGA-FITC revealed chitinous spherical structures in the exopods of the swimming legs in *O. nana* males and females and in *O. similis* females, which were not observed in previous studies. The absence of these structures in *O. similis* male individuals may be a bias due to their low presence in our samples. Despite, luminescence is not conspicuous in *Oithona*, these structures could be luminous glands (*Herring, 1988*). The green staining revealed also a chitinous tubular structure in the exopods penultimate segment of the swimming legs. These structures resemble the 'Crusalis organ,' an osmoregulatory structure that was described by *Johnson et al. (2014)* from the coastal/estuarine copepod *Eurytemora affinis*. Since the globular and the tubular structures seemed attached, we suggest they form only one organ involved either in bioluminescence, osmoregulation or both.

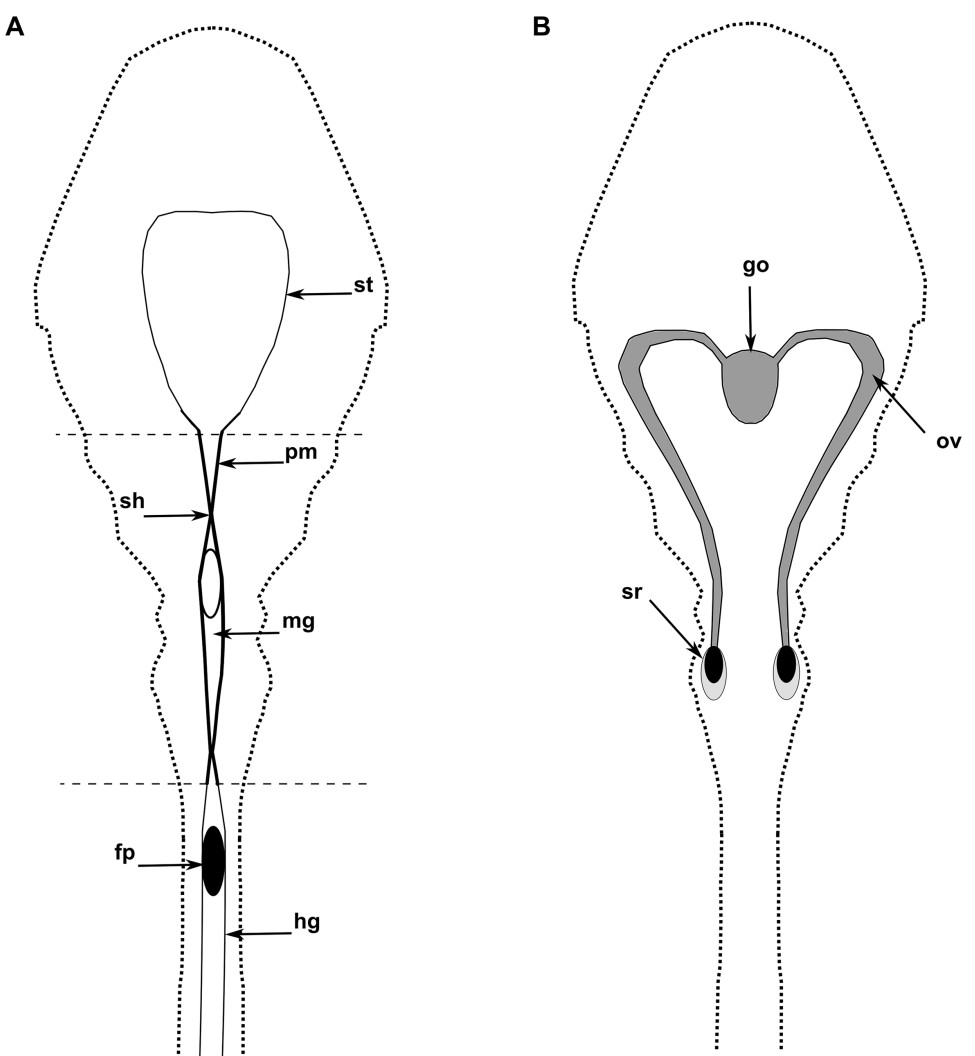

**Figure 5 The diagram of the internal anatomy of a female *O. nana*.** (A) Diagram of the dorsal view of the digestive system. (B) Diagram of the dorsal view of the reproductive system. st, stomach; mg, midgut; pm, peritrophic membrane; go, gonads; sh, shrinkage; fp, faecal pellet; hg, hindgut; sr, seminal receptacle; ov, oviduct. Thick black zones correspond to chitin rich areas. Dark grey zones correspond to heterogeneous chitin area. Light grey zones correspond to amorphous chitin areas.

The WGA-FITC staining allowed also the identification of the chitin distribution in the *Oithona* organs, which provides a high-quality view of the external and internal anatomy and pointed out the major role of chitin in the *Oithona* digestion and reproduction. According to insect studies, the distribution of chitin in the digestive system is limited to the midgut (*Hegedus et al., 2009*; *Terra, 2001*). The same chitin distribution was observed in decapods (*Martin et al., 2006*; *Wang et al., 2012*). In both *Oithona* species, we detected chitin throughout the digestive system, which distinguishes it from insects and decapods, but seems consistent with observations in other free-living cyclopoids made by *Yoshikoshi & Kô (1988)*.

In the PM of some insects (*Hegedus et al., 2009*; *Kelkenberg et al., 2015*; *Lehane, 1997*), chitin plays a role in protection (chemical, mechanical and against viruses, bacteria and pathogens) and digestion (*Terra, 2001*). As the synthesis of chitin has a significant metabolic cost for the organism, we hypothesized that, like the insects and decapods PM, the formation of a chitin coat around faecal pellets help to protect against toxins and pathogens that were not degraded during digestion.

In copepods, no evidence of midgut contraction has previously been described although the phenomenon has been suggested at several instances (*Gauld, 1957*). We suppose that the midgut shrinkages observed in this study could play a key role in the formation and motion of the faecal pellets to the anus. However, we observed intestine shrinkages without the presence of faecal pellets, and vice versa. As proposed by Yoshikoshi and Kô for other copepods, we also suggest that, in *Oithona*, the formation of chitin coat around the faecal pellets can be produced by engulfing digested food in chitin microfibrils present in the PM of the anterior midgut (Fig. 5; *Yoshikoshi & Kô, 1988*).

The presence of chitin along the oviduct and spermiduct walls validates the cuticular appearance of the ducts described by *Cuoc et al. (1997)*. In all *Oithona* males, we observed a pair of spermiducts, while in *Calanus finmarchicus* one of the two spermiducts disappeared during the male differentiation (*Tande & Hopkins, 1981*). The bipartite structure of the seminal receptacles and vesicles found in *O. nana* and *O. similis* males and females were very similar. In males, we hypothesized that the chitinous structure of the vesicle plays a role in the holding of the spermatophores during their formation. Likewise, in the females, this structure would play a role in the holding of the ovisac but also in the opening and closing of the oviduct to release oocytes in the seminal receptacle.

## CONCLUSION

With this study, we adapted and tested a simple and rapid chitin-staining protocol that can help to the taxonomic identification of copepods, and enable new studies on copepod comparative anatomy at a larger scale. The application of the method to *Oithona* extended the knowledge of the structure of its digestive and reproductive systems. Considering the important role of copepods in the carbon and nitrogen sequestration through chitin synthesis, more efforts should be undergone to better understand the molecular and physiological mechanisms involved in faecal pellets formation.

## ACKNOWLEDGEMENTS

We thank Julie Poulain for initiating the *Oithona* genome project and Dr. Leocadio Blanco-Bercial for helpful comments on the manuscript.

### Funding

This work was supported by the Commissariat à l'Energie Atomique et aux Energies Alternatives, the French Ministry of Research and OCEANOMICS (ANR-11-BTBR-0008).

The funders had no role in study design, data collection and analysis, decision to publish, or preparation of the manuscript.

## Grant Disclosures
The following grant information was disclosed by the authors:
French Ministry of Research and OCEANOMICS: ANR-11-BTBR-0008.

## Competing Interests
The authors declare that they have no competing interests.

## Author Contributions
- Kevin Sugier performed the experiments, analysed the data, contributed reagents/ materials/analysis tools, prepared figures and/or tables, approved the final draft.
- Benoit Vacherie conceived and designed the experiments, performed the experiments, contributed reagents/materials/analysis tools, approved the final draft.
- Astrid Cornils authored or reviewed drafts of the paper, approved the final draft.
- Patrick Wincker authored or reviewed drafts of the paper, approved the final draft.
- Jean-Louis Jamet contributed reagents/materials/analysis tools, authored or reviewed drafts of the paper, approved the final draft.
- Mohammed-Amin Madoui conceived and designed the experiments, performed the experiments, analysed the data, contributed reagents/materials/analysis tools, prepared figures and/or tables, authored or reviewed drafts of the paper, approved the final draft.

## Data Availability
  The raw data are provided in the article as photomicrographs.

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
