# Peer review of "Chitin distribution in the Oithona digestive and reproductive systems revealed by fluorescence microscopy"

_PeerJ, doi:10.7717/peerj.4685_

## Round 0.1 · original submission · Major Revisions

I am sorry for the delay, we have been trying to reach the editor who handled your submission without success, so I have taken over to process your submission rather than delay you any longer.

We have reviews back from 2 referees who differ in their opinion regarding your paper. The second referee has few comments and recommends the paper proceed with minor revisions, but the first referee has a couple of major concerns that seem valid to my reading as well. The first criticism is regarding the literature cited, which they feel is insufficient and argue the authors should first review the approaches used in previous studies to highlight the advantage of the proposed double staining method here. For myself, I would have found this highly useful in reading the manuscript. The second criticism is that they do not see the advantage of the double staining and believe that the results could be seen equally well with only the FITC stain, so wonder at the rationale and advantage of subsequent staining with DAPI. I have to say that after reading the manuscript, I am left with the same question, because you do a good job of explaining the rationale and advantage of WGA-FITC, but DAPI seems to just follow along without much explanation. Given that, I find myself closer to the first referee than the second in their evaluation of the manuscript.

Overall, I would characterize both referees as basically supportive of the manuscript, but with some overlapping concerns. I expect that a suitably revised manuscript that addresses the issues raised in the reviews may become acceptable to the reviewers, and I look forward to seeing your revised manuscript.

·

Basic reporting

- In general, the manuscript is nice to read, however, there are some minor language issues. Therefore, the assistance of a native English speaker for proof reading would be helpful.

- The current state of anatomy research of copepods is described insufficiently and the background should be significantly refined.
Particularly, since this study focuses on development of the new staining method, it is especially important to compare it with other staining techniques used for studying digestive and reproductive systems of copepods. However, references to many approaches, which were used in previous studies are lacking, e.g. to gonad observations of formalin preserved individuals (without staining), staining with the borax carmine, fast green, molecular fluorescent probes, gut fluorescence analysis, etc. (e.g. Tande & Hopkins, 1981; Batchelder, 1986; Wolfram et al., 2014; Niehoff & Runge, 2003). So I strongly recommend to briefly review these methods in the introduction and point out the advantages/disadvantages of the proposed double staining (DAPI+FITC) in the discussion section.

- The ms structure is clear. I appreciated the high figures quality, which is usually difficult to get when studying such small organisms. One minor omission – the absence of explanation of the abbreviation (“go” – gonad?) in the legend to the Fig. 4.

Experimental design

There are two important points related to the methodology, which substantially decrease the value of this study:

- The results of the study did not convince me that the double staining with FITC and DAPI is the appropriate method for the study purpose.
To my opinion, figures 1-3 indicate that DAPI staining did not reveal more details of anatomy of the reproductive system than FITC staining (except of the presence of semen). The ovary, oviducts and seminal receptacles are well seen after FITC staining, so additional DAPI staining seems to be excessive. However, my conclusion is based on image analysis, maybe under the microscope it looks different. Nevertheless, in the present form the advantages of staining with DAPI are not obvious for readers and it should be either carefully justified in the text or the findings should be considerably re-considered.
The need for using FITC is well explained in the Introduction (lines 33-40), but the reasons for subsequent DAPI staining are not discussed there (only briefly in Methods, line 57), so I suggest to insert this information to indicate clearly, whether DAPI staining is really necessary. DAPI is a non-specific fluorochrome causing fluorescence of all tissues and it is not obvious from the text why it has been chosen to study the reproductive system instead of, say, lipid-specific stains.

- The quantity of material, which has been analyzed, is not clear enough.
The number of samples should be indicated as well as numbers of individuals stained (for both Oithona species) and their age (copepodite stages/adults).
Please, clarify also, why such sampling periods (November 2016, January 2017, March 2017) was chosen. To my knowledge, copepods usually do not reproduce actively at low temperatures and these months do not seem to be well suited for studying their reproductive system (in the peak of reproductive season much more gonad stages are available).

Validity of the findings

- Some generalizations in Discussion are not supported by references and the presented results.
For example, one of the findings discussed (lines 129-130) is that the presence of chitin throughout the digestive system, distinguishes Oithona from other arthropods. However the authors provided references only about decapods and insects for comparison (lines 128-129), whereas phylum Arthropoda includes also many others taxonomic groups (e.g. subphylum Chelicerata, Myriapoda). So the statement about “other arthropods” (line 130) should be removed or supported by appropriate literature data.
In addition, in the Introduction it was mentioned (lines 28-30) that some free-living copepods contain chitin in the anterior and posterior midgut peritrophic membrane (PM) according to previous studies. Therefore, if I understand properly, the chitin localization is similar to observed in Oithona (throughout the digestive system)? If so, this finding (lines 129-130) should be revised.

- The description of some results is not sufficiently detailed:

1. It is reported that chitinous elliptical structures of unknown function were observed AT the exopods (line 81) and IN the exopods (line 122). Therefore, it is unclear whether they are internal structures or localized at the surface. In addition, please indicate were these chitinous structures observed only in O. nana females (and what is about O. similis)? Did they present in males, copepodits and nauplii? Besides of luminous glands (line 123) these structures could be for example, a result of moulting. Anyway, more detailed data about these structures should be provided to make assumptions.

2. Since two related copepod species were studied, it would be good to discuss similarity/dissimilarity in their internal anatomy separately. This will be a valuable addition to the discussion.

3. It is unclear what hypothesis was tested by analysis of correlation between the presence of intestine shrinkages and fecal pellets (lines 139-140). If the shrinkages are temporal structures visible only at the moment of gut contraction, the absence of shrinkages does not necessarily reflect that they are not involved in the pellet motion. Therefore, the correlation analysis seems to be useless here. In addition, the fact that some observed individuals did not have shrinkages is not indicated in the results (lines 94-97). Please, clarify also whether there were copepods with shrinkages and without pellets and vice versa (without shrinkages and with pellets), if you discuss this item.

Additional comments

Dear Authors,

I found your manuscript about anatomy of important but yet understudied Oithona species interesting, but requiring a serious revision. The proposed WGA-FITC staining procedure gives very nice results and seems to be the helpful tool for studying digestive and reproductive systems of copepods. However the rationality of subsequent DAPI staining causes some doubts (please, see my detailed comments). Unfortunately, the gaps in explaining the relevance of the new staining method, serious inaccuracies in literature review and discussion listed in my comments reduce the potential impact of this study, thus I would recommend an additional round of careful editing. Hopefully, my comments will help you to improve the manuscript.

Best wishes,
Katya

·

Basic reporting

No comment

Experimental design

The authors write that the samples of zooplankton after collection were immediately placed in ethanol (line 50-51). This indicates that the procedure for defecation of the copepod intestine was not performed. The question is «why»?
The Copepod are filters and can consume food from a water that differs in composition and genesis. The composition of which may include marine fungi plankton and detritus containing chitin. The authors write "In both Oithona species, we detected chitin throughout the digestive system, which distinguishes it from other arthropods." (line 129-130). Probably, this coloring gave not only chitin lining the walls of the intestine, but also chitin, contained in the food of crustaceans. I would like to clarify this point. Why the authors in the discussion do not consider the possibility that the full intestine of the crustaceans has distorted the data on the distribution of chitin in the digestive tract of copepods.

Validity of the findings

No comment

Additional comments

line 154-155
"The application of the method to Oithona extended the knowledge of the biochemistry and structure of its digestive and reproductive systems".

replace with

"The application of the method to Oithona extended the knowledge of the structure of its digestive and reproductive systems".

because the biochemical structure in this work was not considered. In fact, the article is devoted to the morphology digestive and reproductive systems and methods of their investigation.

---

## Round 0.2 · Minor Revisions

Thank you for your revised submission, which the more critical referee denotes is substantially improved. However, I find that I concur with their remaining criticism that your response to the issues raised by the referees needs to be incorporated into the paper itself, and not simply in the rebuttal letter. I agree with the referee that incorporating the specific conditions under which FITC staining is sufficient and those under which DAPI staining provides additional benefit should be outlined in results and discussion of the paper. The referee appears satisfied with your response, but expects that the changes be made to the body of the manuscript so that the next reader will be aware of these issues as well, rather than hiding them in the response to reviewer's comments. I find myself in agreement with this position, and am returning the manuscript to you for minor revisions to incorporate this change. Once this issue is resolved, I do not expect the paper will require additional review.

·

Basic reporting

General comments:

The ms has been substantially improved, however some important points are still not fully addressed. As I understand, the authors agree that the proposed subsequent DAPI staining provides only minor additional information to the results of FITC staining and seems to be excessive when studying structure of digestive system and gonads. The double-staining is rational for addressing the narrow range of tasks (e.g. to check the presence of semen). So, it is necessary to focus on the appropriateness of the proposed double-staining method (particularly, DAPI step) in results and discussion (not only in the response to reviewer's comments). I can't find this information in the text, however it is one of the main results of this study.
I also have several minor comments concerning the introduction section (some parts should be reformulated), methods and results, please find them below.

Experimental design

Unfortunately, the line numbers indicated in the authors' response do not correspond to the actual changes...

Introduction

The paragraph about methods of studying copepods is still confusing and difficult to read, reformulate it, please:

Lines 58-61: It could be compiled since electronic microscopy has been widely used for studying different copepods (not only calanoids and Oithona). It is also important to indicate that these methods are additional to light microscopy observations. For example: “The details of external anatomy of copepods has been also analyzed through Congo red fluorescence [18] and electronic microscopy [19], [2].”

Lines 61-64: Should be shortened (formalin is an aldehyde, so “or” could not be used here; only cell, but not gonad can have nucleus and cytoplasm). I suggest something like: “Reproductive system of copepods preserved with aldehydes has been studied by direct observation of individuals [20-23], staining of gonads with borax carmine [22, 24], fluorescent polyunsaturated aldehydes (PUAs) probes [25], or with Fast Green [26].”

Lines 67-69: Please, re-formulate this sentence so that it would begin: “The anatomy of digestive system was studied using...

Lines 74-75: better to move this sentence to the beginning of paragraph (and make it the 2nd). If the mentioned diagrams are based on light microscopy observations it is necessary to notice.

Methods

Line 99: Although the authors consider impossible to provide information about the number of samples and stained individuals, at least data about the stages/sexes of stained copepods are needed here. Please, clarify this before the description of staining procedures.

Line 106: delete the point after g.

Validity of the findings

Results

It is very important to give the general information about the effectiveness of the proposed staining method in the beginning of results, before the describing of detected structures. Please, summarize it also in the beginning of the discussion.

Line 152: as I understand, not only DAPI, but also FITC staining makes the gonads visible.
Line 154: Replace Mironova et al. by Mironova & Pasternak

Conclusions

Line 231-233: it is too specific issue for conclusions, to my opinion, better to delete this sentence.

---

## Round 0.3 · accepted · Accept

Thank you for clarifying the manuscript with respect the the specific advantage of the double-staining, and the insights gained by DAPI which were not possible with traditional FITC staining. I am satisfied with your revised text, and am happy to move your manuscript forward into production.

#